# Numerous rRNA molecules form the apicomplexan mitoribosome via repurposed protein and RNA elements

Shikha Shikha [1,2,5], Victor Tobiasson [3,5], Mariana Ferreira Silva[1,2], Jana Ovciarikova[1,2], Dario Beraldi [1], Alexander Mühleip [1,2,4] & Lilach Sheiner [1,2]

Mitochondrial ribosomes (mitoribosomes) are essential, and their function of synthesising mitochondrial proteins is universal. The core of almost all mitoribosomes is formed from a small number of long and self-folding rRNA molecules. In contrast, the mitoribosome of the apicomplexan parasite *Toxoplasma gondii* assembles from over 50 extremely short rRNA molecules. Here, we use cryo-EM to discover the features that enable this unusual mitoribosome to perform its function. We reveal that poly-A tails added to rRNA molecules are integrated into the ribosome, and we demonstrate their essentiality for mitoribosome formation and for parasite survival. This is a distinct function for poly-A tails, which are otherwise known primarily as stabilisers of messenger RNAs. Furthermore, while ribosomes typically consist of unique rRNA sequences, here nine sequences are used twice, each copy integrated in a different mitoribosome domain, revealing one of the mechanisms enabling the extreme mitochondrial genome reduction characteristic to Apicomplexa and to a large group of related microbial eukaryotes. Finally, several transcription factor-like proteins are repurposed to compensate for reduced or lost critical ribosomal domains, including members of the ApiAP2 family thus far considered to be DNA-binding transcription factors.

Mitochondrial ribosomes (mitoribosomes) play the central and ubiquitous role of synthesising mitochondrial proteins which are essential for the survival of almost all eukaryotes including parasites. Accordingly, apicomplexan parasites which cause deadly human and animal diseases such as toxoplasmosis and malaria require their mitoribosome for survival[1–3] and mitoribosomal inhibition is considered a promising avenue for drug discovery[4–7].

Ribosomes, including mitoribosomes, rely on a small number, most commonly two and up to 16[8–11], of ribosomal RNA (rRNA) molecules to perform their essential and universal function. Mitoribosomal

RNAs are encoded by the mitochondrial genome (mt-genome), typically by long continuous genes that enable the co-translational rRNA folding necessary to form critical ribosome functional domains. However, a growing number of studies report mt-genome sequences where the typical small number of long and continuous rRNA coding genes are 'fragmented' into many short sequences[12–16]. In the apicomplexans, the mt-genome encodes tens of extremely short rRNA molecules. It is not understood how mitoribosomes can still form or perform their function when their rRNA is transcribed in many short fragments, nor what adaptations evolved to support the continued

¹School of Infection and Immunity, University of Glasgow, Glasgow, Scotland, UK. ²Glasgow Centre for Parasitology, University of Glasgow, Glasgow, Scotland, UK. ³National Center for Biotechnology Information, National Library of Medicine, Bethesda, MD, USA. ⁴Institute of Biotechnology, Helsinki Institute of Life Science HiLIFE, University of Helsinki, Helsinki, Finland. ⁵These authors contributed equally: Shikha Shikha, Victor Tobiasson. ✉e-mail: alexander.muhleip@helsinki.fi; lilach.sheiner@glasgow.ac.uk

function of the mitoribosome given this fragmentation. To answer these questions, we turned to study the mitoribosome which is predicted to have both the highest number and the shortest rRNA molecules known in nature[12,16]: that of the widespread apicomplexan human parasite *Toxoplasma gondii*. We asked what proteins and RNAs were recruited, and how conserved proteins and ribosomal domains were remodelled, to maintain the universal ribosome function essential for these parasites' survival and ability to cause disease.

The *T. gondii* mt-genome has an unusual architecture[12,16]. This mt-genome is composed of numerous copies of conserved and repetitive sequence blocks that are organised on multiple DNA molecules. The blocks are arranged end to end forming DNA molecules of varying lengths and with varying block combinations[12,16]. We hypothesised that this unusual arrangement of genetic elements would have a major impact on mitoribosome evolution, a hypothesis that is supported by the prediction of no less than 36 short rRNA molecules encoded on this mt-genome[12,16]. High numbers of short rRNA molecules are similarly predicted in the mt-genome of the closely related malaria-causing *Plasmodium* spp[12,17,18] and in the mt-genomes of a broad and divergent group of related eukaryotic microbes[13–15]. While evidence for *Toxoplasma* and *Plasmodium* mitoribosome assembly and activity is accumulating[1–3], the structural and functional implications of this so-called 'rRNA fragmentation' are unknown. Here we show the full magnitude of rRNA fragmentation in the model apicomplexan *T. gondii* and the extensive adaptations acquired to maintain a functional mitoribosome.

## Results

*T. gondii* mitoribosomes were purified through immuno-precipitation via the endogenously FLAG-tagged large ribosomal subunit protein bL12m (Supplementary Fig. 1, Supplementary Data 1) and the structure determined by cryoEM (Supplementary Fig. 2),. We resolved the complete mitoribosome structure with map resolutions of the large and small ribosomal subunits (LSU and SSU) ranging from 2.2 to 2.8 Å (Fig. 1, Supplementary Fig. 2, Supplementary Table 1). While the sequence and the local structures of the decoding and peptidyl transferase centres are conserved with the bacterial ribosome, the overall ribosome architecture is distinct from the bacterial ancestor of mitoribosomes, and from any mitoribosome described to date. The structure consists of a total of 124 modelled proteins, as well as a $Zn^{2+}$, a $Mg^{2+}$ and an ATP, molecules in the SSU and a $Zn^{2+}$ and two $Fe_2S_2$ in the LSU. Among the proteins 55 are clade-specific (Supplementary Data 2,

Supplementary Data 3), forming a proteinaceous layer around the heavily fragmented rRNA core. *T. gondii* and the apicomplexan phylum belongs to a broader group of eukaryotic microbes, named Myzozoa, that further includes marine parasites, free-living algae and coral-symbionts. Myzozoans share the feature of numerous and small mitochondrially encoded rRNA molecules[13–15], and accordingly, the clade-specific proteins found in the *T. gondii* mitoribosome include 44 myzozoan conserved proteins (Supplementary Fig. 3), suggesting co-evolution of rRNA and proteins. Among these clade-specific proteins we found numerous cases of heterodimers, multi-copy proteins and protein-pairs with identical or closely related sequences or folds which we named collectively twin-elements (Fig. 1, Supplementary Fig. 4). Mitoribosomal proteins are typically single-copy, and structurally distinct from each other. While cases of twin-elements do exist in other mitoribosomes, their high occurrence here might be a signature feature of the *T. gondii* mitoribosome, and their conservation (Supplementary Fig. 3) suggests that this might apply across the Myzozoa.

**Despite the fragmentation into 53 small molecules, the *T. gondii* mitoribosome has a large rRNA core, that includes re-used rRNA molecules**

To accurately assess the extent of rRNA fragmentation in the *T. gondii* mitoribosome, sequences were first identified directly from the EM density, referenced against the mt-genome blocks[12]. We then further cross referenced the sequences against RNA-sequencing data that we generated from the immunoprecipitated sample (Supplementary Data 4, Supplementary Fig. 5, Supplementary Fig. 6). The result of these analyses reveals a total of 53 rRNA molecules (Fig. 2), markedly exceeding the previously predicted level of fragmentation[12,16]. The 32 LSU and 21 SSU rRNA molecules are extremely short, ranging from 278 to just seven nucleotides. We labelled these fragments corresponding to subunit and numbered relative to the topology of the bacterial ancestor, LSU-1 to LSU-32 and SSU-1 to SSU-21. A comparison to previously studied mitochondrial[8,9] and cytosolic[10,11] ribosomes that consist of far fewer rRNA molecules, found common points of fragmentation (Supplementary Fig. 7) highlighting a general convergence of the fragmentation process and suggesting structural constraints of the fragmentation process across lineages. In addition to the number and lengths of the rRNA molecules, we also observed several cases of rRNA permutations and more complex topological rearrangements, as reported previously for mitoribosomes with

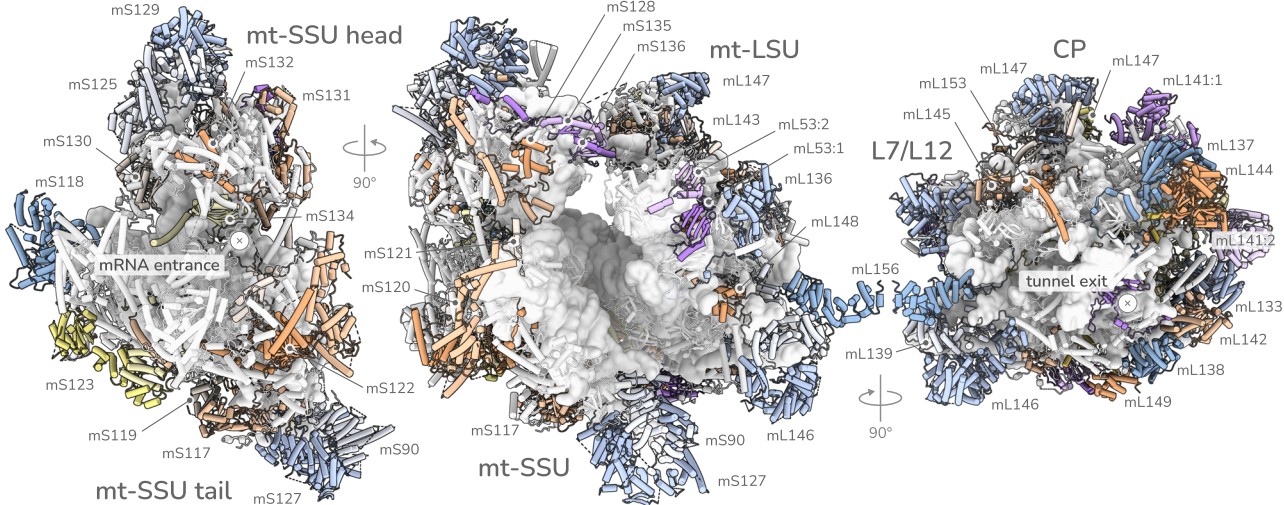

**Fig. 1 | The mitochondrial ribosome from *T. gondii*.** CryoEM reconstruction of the complete *T. gondii* mitoribosome. Central panel showing monosome, left and right panels showing separate small subunit (SSU) and large subunit (LSU) respectively. The rRNA is shown as a white surface representation. Clade-specific proteins are shown as coloured cartoons. Proteins with helical repeat elements in blue; 'twin-elements' including heterodimers, and duplicate proteins in purple; remaining clade-specific proteins in shades of orange and yellow.

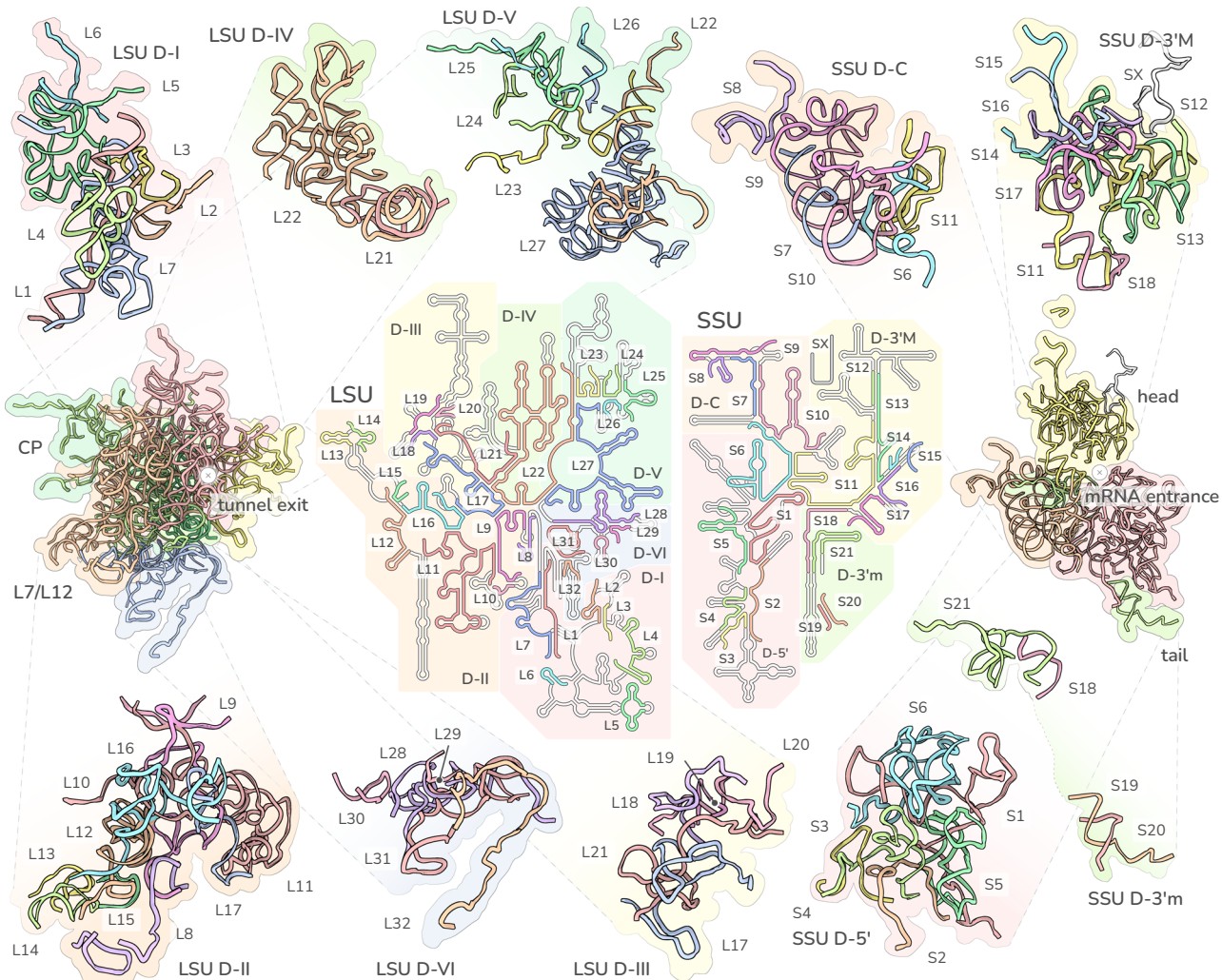

**Fig. 2 | Overview of rRNA fragment arrangement in 2D schematic and 3D structure.** 2D topological diagram shown in the centre, coloured serially by fragment number, and overlaid onto the 2D structure of the *E. coli* ribosomal rRNA (grey). Cartoons of rRNA 3D structure to the left and right show the full LSU and SSU respectively, coloured by rRNA domain. Top and bottom panels show the 3D structure of individual rRNA domains coloured by fragment number.

fragmented rRNA molecules[8,9]. Finally, we observed 10 cases of adenine to guanine (A → G) transitions between the identified rRNA and the genomic sequence[12]. These changes are strongly supported by local density (Supplementary Fig. 8) and observed also in reads from our rRNA-sequencing data (Supplementary Data 4). These transitions may result from heterogeneity within mitochondrial genome blocks[17] or via targeted rRNA editing by a thus far uncharacterised machinery[18]. However, our rRNA-sequencing data and density support a model where only a minor pool of the rRNA transcripts exhibiting the A → G transition (1–20% depending on which rRNA) preferentially gets incorporated into the mature ribosome.

Despite the high degree of rRNA fragmentation, the identified rRNA molecules assemble into an overall rRNA core of ~1780 nts for the LSU and ~1020 nts for the SSU, larger than several previously reported structures[8,19,20] and significantly larger than predicted for *Toxoplasma* or *Plasmodium*[12,21,22]. In line with this unexpectedly large size, we also observed seven rRNA expansion segments, three in the LSU and four in the SSU (Fig. 2, Supplementary Fig. 6), where conserved sequences have insertions compared to the bacterial ancestor. The simultaneous observation of a relatively large rRNA core alongside such extensive fragmentation highlights a previously unseen independence of the two evolutionary drivers of ribosome divergence: rRNA reduction and fragmentation. As such, the *T. gondii* mitoribosome refutes the concept of fragmentation occurring merely as a side-product or a possible driver of rRNA reduction and highlights fragmentation as a separate, independent, driver for ribosome evolution.

Strikingly, when mapping the rRNA sequences against the mt-genome blocks (Supplementary Fig. 5) we discovered nine cases where the same sequence is 're-used' in two different places in the mitoribosome (Supplementary Data 4, Supplementary Fig. 5). In all cases one of the re-used copies adopts its ancestral structure, as observed in the bacterial core, while the other is observed with little higher order structure (Fig. 2, Supplementary Fig. 5). These re-used sequences, which range in length from 14 to 46 nucleotides, are linked to the order of the mt-genome block arrangement (Supplementary Fig. 5). We concluded that the *T. gondii* mitoribosomes are 'forced' to assemble from sequences available following the reduction and scrambling that the mt-genome has undergone during the evolution of these parasites, providing an extreme example for the evolutionary malleability of rRNA.

**A mitochondrial polyadenine polymerase is essential for parasite survival and rRNA polyadenylation is necessary for mitoribosome stability**

A second distinct characteristic of the *T. gondii* mitoribosome is the post-transcriptional addition of 3′ poly-adenosine (poly-A) tails to

the rRNA molecules, incorporated as well-defined structural elements in all but 11 rRNAs. Polyadenylation of rRNA has been reported in a handful of studies[23,24] however its function has remained elusive. Here we observed these poly-A tails directly in the mitoribosomal rRNA core. Interestingly, due to their short length, most rRNA molecules in the *T. gondii* mitoribosome are likely unable to associate or fold independently to form the ribosomal functional core. This is in contrast to all previous studied ribosomes, including those with fragmented rRNA, which consist of fewer and larger rRNA molecules, capable of forming extensive inter- or intra-molecule base pairing and of the formation of solvent-excluded interfaces. This lack of rRNA molecule interactions in *T. gondii* undoubtedly poses a challenge for ribosome assembly. We hypothesise that the poly-A tails serve as 'handles' that are bound by proteins which, in turn, interact with the nascent ribosome to facilitate assembly and promote stability (Fig. 3A left, Supplementary Fig. 9). In support of this hypothesis, the 11 fragments that do not have poly-A tails are either large enough to form well-defined domains (e.g. LSU-11, LSU-17, SSU-10, SSU-11) likely capable of independent assembly; or appear unable to harbour poly-A tails due to steric constraints from surrounding rRNA or protein (e.g. LSU-5, LSU-6, SSU-13). To assess this hypothesis, we identified a poly-A polymerase homologue encoded by the gene TGGT1_281370 (Supplementary Fig. 10) with both strong evidence for mitochondrial localisation[25] and a high essentiality score[26], which we here name TgmtPAP1. We then generated a conditional depletion mutant of TgmtPAP1 (Supplementary Fig. 10). In line with its essential role, depletion of this poly-A polymerase led to a severe growth defect (Fig. 3B, C). RNA sequencing confirmed that upon TgmtPAP1 depletion several rRNA molecules manifested shorter poly-A tails compared to wild-type parasites (Supplementary Fig. 9), however the tails were not fully ablated suggesting potential residual activity or the presence of another polymerase. We analysed the impact of the resulting shortened tails on the mitoribosome. To assess mitochondrial translation we used our indirect translation assay[2,3]. This assay compares the signals from fully assembled complex IV, for which two components are encoded within the mitochondrial genome (coxI, coxIII) and thus requires mitochondrial translation, to the signal from fully assembled complex V, for which all components are translated by cytosolic ribosomes. The mitochondrial genome also encodes one subunit of complex III (cob) however we did not examine its assembly and rather used only complex IV as reference, as we did previously[2,3]. In support of our hypothesis of the role of poly-A tails in mitoribosome formation or stability, depletion of TgmtPAP1 led to a mitochondrial translation defect[2,3] (Fig. 3E, Supplementary Fig. 11). However, while traditional mRNA polyadenylation in *T. gondii* is not strongly supported[12] it cannot be excluded that the observed translation defect may be due to mRNA polyadenylation rather than mitoribosome formation defect. Thus, we directly analysed the native gel migration of bL12m-FLAG upon TgmtPAP1 depletion (Fig. 3F), where we observed reduction of the high band corresponding to the mitoribosome, while the expression level of bL12m-FLAG remains largely unchanged, indicative of a mitoribosome formation or stability defect upon *TgmtPAP1* depletion. Other mitochondrial functions not dependent on mitochondrial translation such as the assembly of the mitochondrial protein import translocon, TOM and the enzymatic activity of the mitochondrial electron transport chain complex, SDH, are unaffected (Fig. 3D, Supplementary Fig. 10), supporting specificity. We found that TgmtPAP1 is highly conserved among apicomplexans (Supplementary Fig. 10) suggesting that polyadenylation of rRNA assisting mitoribosome assembly or stability may be a conserved trait. In support of this possibility, rRNA with poly-A tails have also been detected in *Plasmodium*[15,27].

## RNA binding proteins and numerous protein extensions mitigate the extreme rRNA fragmentation by maintaining rRNA fragment association

Another potential solution to the assembly challenge may be presented via proteins mediating interactions between multiple rRNA termini. We observed 11 proteins that likely interact with more than two rRNA molecules (Supplementary Data 2). We hypothesise that these interactions help colocalise rRNA molecules during assembly, thereby serving as joints between them. One example, mL138, is closely associated with two rRNA molecules (LSU-20 and LSU-21) and interact with four more (LSU-11, LSU-17, LSU-18, LSU-19) (Fig. 3A right). Following our hypothesis, these rRNA interactions are predicted to be critical for mitoribosome formation. To examine this possibility, we made a conditional depletion mutant of mL138 (Supplementary Fig. 10). In support of an essential role in mitoribosome formation, mL138 depletion resulted in mitoribosome assembly or stability defect, leading to a mitochondrial translation defect and ultimately to parasite death (Fig. 3B–F, Supplementary Fig. 11). mL138 is one of five RAP (RNA-binding domain abundant in apicomplexans) domain-containing proteins found in the *T. gondii* mitoribosome (Supplementary Data 2), an observation which is in line with a suggested role of RAP proteins in mitoribosome regulation in the related *Plasmodium*[28]. Interestingly, while RAP proteins are diverse in sequence and domain composition, mL138 is strictly conserved across Myzozoa (Supplementary Fig. 3), suggesting an ancestral and potentially conserved role of rRNA-molecule mediation in mitoribosomal biogenesis.

Finally, we observe an extensive incorporation of low complexity protein elements mediating local rRNA-rRNA interactions. These include a set of short peptides (<40 amino-acid long), six in the LSU and two in the SSU (Supplementary Data 5), heavily enriched in aromatic residues capable of stacking with nucleobases and charged residues able to interact with backbone rRNA, both promoting association between rRNA molecules (one example in Fig. 3A bottom). This 'charge-compensating' role was previously described for other mitoribosomal peptides, such as the universal mS38, the ciliate specific mS75 and the algal mL128[8,29], however the abundance of such peptides is notably increased in the *T. gondii* mitoribosome (eight here compared to individual cases seen in the above mentioned systems). In addition to peptides, we found charge-compensating motifs also in C- and N-terminal protein extensions of universally conserved mitoribosomal proteins. Despite being a hallmark of mitoribosome evolution, the extensions of conserved mitoribosomal proteins and their role in mitoribosome biology remains unstudied. One theory, 'constructive neutral evolution'[30], would suggest that upon initial acquisition these extensions may not have provided an immediate fitness advantage, but became essential during subsequent remodelling[29]. Based on the observations here and other mitoribosomes[8–11,29], we postulate that in ribosomes with either fragmented or reduced rRNAs, protein extensions with charge compensating properties gained the function of promoting rRNA-rRNA contacts and rRNA stability.

## Several proteins with transcription factor domains are repurposed to compensate for a reduced subunit interface

Detailed studies in bacterial systems have indicated key positions across the subunit interface essential for ribosome association and translational dynamics[31,32]. In *T. gondii* the subunit interface is highly divergent from the bacterial ancestor, and interactions outside the core helices H69 and h44 are lost or replaced (Fig. 4). The remodelling include reduction and fragmentation of the critical contact mediator rRNA helix 44 (h44)[31]. While the remaining fragments of h44 are scaffolded by helical repeat proteins, they do not directly contribute to subunit interactions (Fig. 4D). Instead, we find two myzozoan conserved LSU proteins with transcription factor-like folds. One of those is

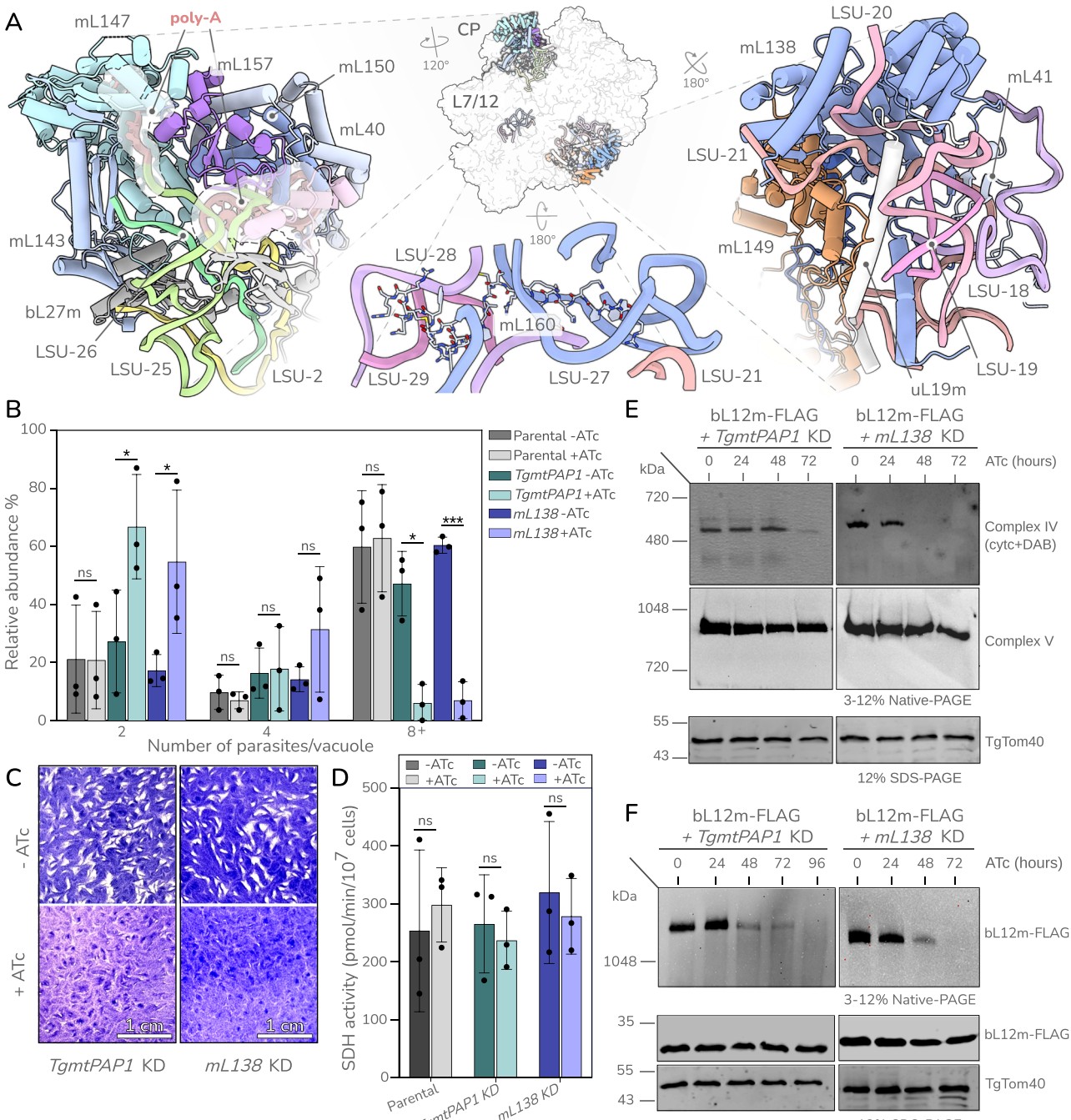

**Fig. 3 | rRNA-protein interactions that stabilise the mitoribosome supporting its function. A** Top left—incorporation of post-transcriptional poly-A tails (red) in fragments LSU24 and LSU25 contribute to protein binding interfaces in the central protuberance (CP), mediated by mL147 in LSU25. Top middle—rRNA-proteins interactions stabilising rRNA fragment contacts shown as cartoon, coloured by protein chain. Bottom middle—an example of charge compensating peptide incorporated into the rRNA core. Right—protein joint mL138 colocalizing termini of LSU-20 and LSU-21 which together with prominent protein extensions from uL19m, mL41 and mL149 help form the reduced LSU domain III. **B**–**E** Phenotypic assays performed with *mL138* and *TgmtPAP1* conditional knockdown (KD) parasites grown in the absence or presence of anhydrotetracycline (ATc) for the indicated time. **B** Replication assay. Mean value (±standard deviation) of number of parasites per vacuole counted in a total of 100 vacuoles per condition for each cell line. *n* = 3 biological replicates. ANOVA (one-sided) followed by Tukey's multiple pairwise comparison (two-sided) between different pairs as indicated. ns not significant, *$p$ < =0.05, ***$p$ < =0.0005. **C** Plaque assays assessing growth of parasites in the host-cell monolayer after 10 days of treatment. Scale bar = 1 cm. **D** SDH activity assay. Mean values with standard deviation are plotted. *n* = 3 biological replicates. Two tailed paired t-test used (ns not significant). **E**. Mitochondrial translation assay. Upper panel shows the mitochondrial translation dependent complex IV assembly and activity; middle panels show the mitochondrial translation independent assembly of Complex V. TgTom40 in the lower panel is used as loading control. **F** Mitoribosome formation assessed via clear native (cn)-PAGE and western blot detecting bL12m-FLAG (upper panel). SDS-PAGE blot in the lower panel shows that the expression of bL12m-FLAG remains the same in this condition. TgTom40 used for loading control. Native- and SDS-PAGE western blot experiments were repeated at least 3 times. Source data are provided as a Source Data file.

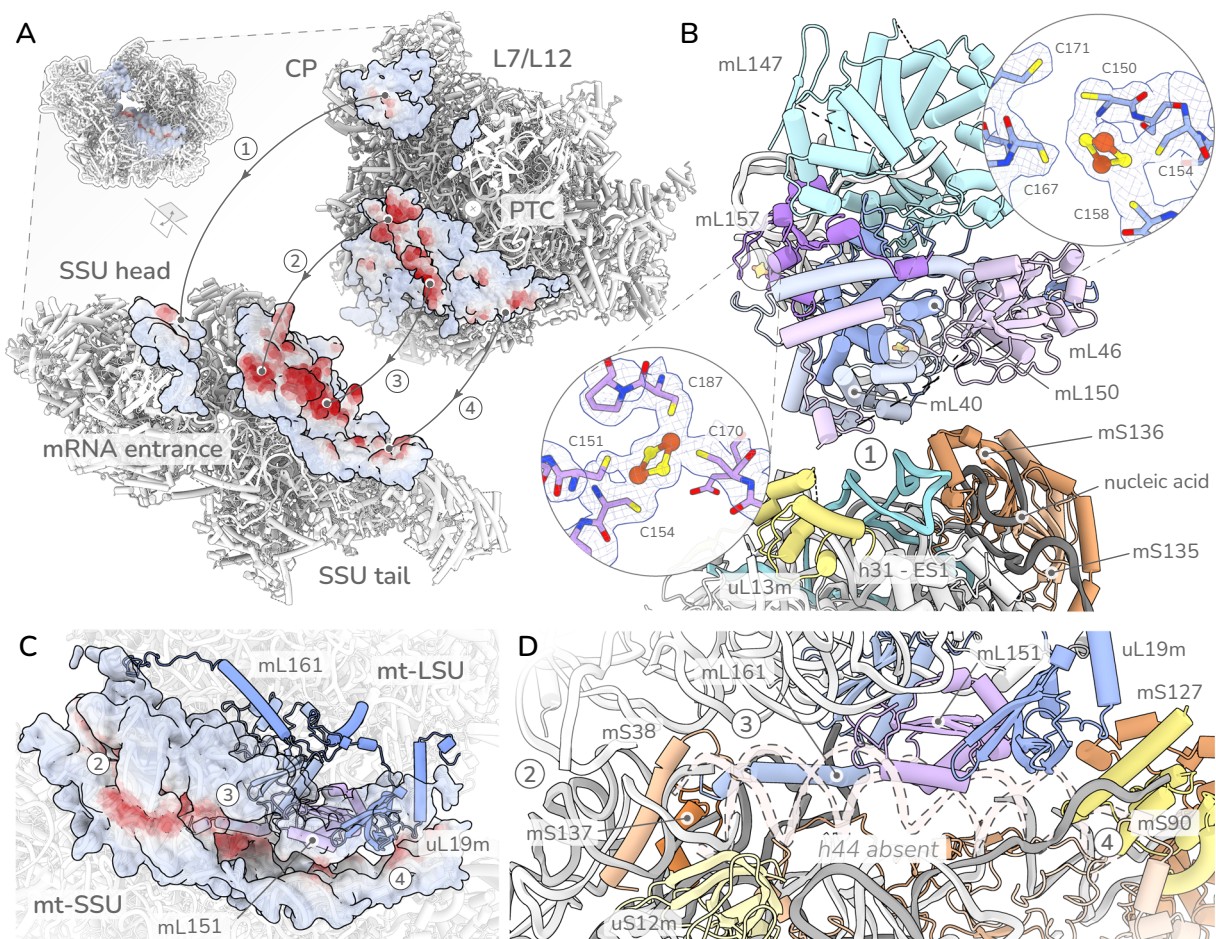

**Fig. 4 | Protein remodelling of subunit interfaces. A** Split view of the subunit interface with points closer than 20 Å shown as surfaces coloured by intersubunit distance (red < 5 Å, blue > 5 Å). Four main contact points numbered as 1: CP/HEAD interactions, 2:H69/h44 contacts, 3: remodelled mL161/SSU regions, 4: SSU-tail/ uL19m surface. **B** Interaction surface 1: Contacts between CP and SSU head mediated by SSU rRNA expansions structured by a mS134/135 heterodimer. CP contacts mediated via expansions from conserved mL40/mL46. Insets showing FeS coordination of mL150 and mL157. All proteins shown as cartoons and coloured individually by chain. **C** Overview of SSU platform interaction surfaces 2, 3 and 4 showing loss of canonical interface mediator h44 as red outline. Contact points 3 and 4 mediated by *Toxoplasma* specific elements. **D** SSU platform contacts. Contact point 3 mediated by C-terminal protein extensions of mL161 bound to the SSU platform by mL38. Contact point 4 mediated by uL19m scaffolded by mL161 interfacing with clade-specific mS90 and mS127. The bacterial ancestor h44 is depicted in a dashed line for comparison. All proteins shown as cartoons and coloured individually by chain.

a protein containing a BolA-like domain, mL151, which interfaces with uL19m. The BolA-like domain was identified via structure-structure comparison tools (foldseq and DALI). The other is a protein containing ApiAP2-like[33,34] domain, mL161, which tethers both subunits via an N-terminal domain embedded in the LSU, and a C-terminal extension that penetrates the SSU platform, running parallel to the reduced h44 (Fig. 4C). Apicomplexan proteins containing ApiAP2 domains were thus far considered to function as transcription factors and this observation raises a function for an ApiAP2 member that was not considered before. To provide support for the role of mL161 in mitoribosome formation we generated a conditional depletion mutant line and demonstrated a mitoribosome formation or stability defect followed by a mitochondrial translation defect and parasite death upon mL161 depletion (Supplementary Fig. 11).

Further subunit interface remodelling is evident in the contacts between the LSU central protuberance (CP) and SSU head. In the bacterial ancestor, interactions in this region are mediated by proteins uL5m, bL31m, uS13m and LSU rRNA H38. In *T. gondii* these elements are either lost (uL5m, bL31, H38) or reduced (uS13m). Subunit association is instead mediated via rRNA through an SSU expansion

segment h31-ES1, and via a protein heterodimer with transcriptional-activator-PC4-like (PC4) domains, mS135 and mS136 (one of two heterodimers seen in this mitoribosome) (Supplementary Fig. 4). The PC4-like heterodimer is positioned on the top of the SSU head, poised for interactions with the CP, and associates tightly with a strand of nucleic acid (Fig. 4B). This strand does not coincide with any known rRNA structure, nor does it appear to be an expansion segment, and is therefore unlikely to be rRNA. As PC4-like heterodimer typically binds DNA, we cannot exclude the possibility that the observed strand is in fact a DNA molecule. Regardless of its identity, its positioning mediates the interaction between the heterodimer and the mitoribosome, and is therefore entrenched as a distinct nucleic acid element of this mitoribosome.

The contact sites on the CP are likewise remodelled, exclusively mediated by protein extensions from the widely conserved proteins, mL40 and mL46 (Fig. 4B). The positioning of this protein module relative to the SSU head is mainly conserved across eukaryotes. However, the underlying interface between mL40 and mL46 with other LSU proteins differs between mitoribosomes. For example, in bacteria-like mitoribosomes, such as in *Arabidopsis thaliana*, this interaction is

primarily mediated via the universal protein uL5m[35], which is lost in *T. gondii*. In its place we instead observe two proteins, mL150 and mL157, which provide the binding interface for mL40 and mL46 (Fig. 4B). mL150 and mL157 therefore act as a keystone in maintaining LSU-SSU contacts. However, neither protein features a substantial tertiary fold. Instead, each protein includes a prominent, solvent accessible, rhombic Fe$_2$S$_2$ cluster (Fig. 4B). FeS clusters were recently suggested to play a structural role during mitoribosome assembly[36]. The positioning of mL150 and mL157, the arrangement of the coordinating cysteines and the solvent accessibility of both FeS clusters, highlight a potential that these proteins may likewise play a role during ribosome assembly or translation. To provide support for the role of mL157 in mitoribosome formation we generated a conditional depletion mutant line and demonstrated that depletion of this gene results in a mitoribosome formation or stability defect along with a mitochondrial translation defect and leading to parasite death (Supplementary Fig. 11).

## Reduction and remodelling of the mRNA channel and exit tunnel

In addition to the extensive remodelling of the subunit interface, we also observe a lineage-specific remodelling of the mRNA entrance and exit regions, and here again, ApiAP2-domain containing proteins were recruited to fill in for reduction and loss of otherwise conserved mitoribosomal proteins. Of the four highly conserved proteins that bind along the canonical mRNA channel found in mitoribosomes of other lineages, only uS5m is present in the *T. gondii* mitoribosome (Fig. 5A). Two proteins are lost (uS4m and uS7m), and one (uS3m) is heavily reduced, retaining only a small fragment corresponding to the ciliate mS92[29]. These structural elements are instead replaced with clade specific proteins and protein extensions. The missing uS3m binding site is occupied by several terminal extensions from surrounding proteins without detectable sequence similarities or structural convergences (uS14m, mS92 and mS134), while uS7m is replaced by a tandem ApiAP2-domain containing protein, mS130 (Fig. 5B, Supplementary Fig. 12). In this sense, the remodelling of the mRNA channel provides a second demonstration of the *T. gondii* mitoribosome signature features of heavy incorporation of protein extensions, proteins with transcription factor-like folds and 'twin-elements' as seen in the remodelled SSU platform, head and CP.

Finally, previous studies of mitoribosomes have observed various alterations to the late ribosomal exit tunnel and joining 'vestibular' area, proposed to be adaptations to the translation of hydrophobic membrane proteins[8,20,37]. As *T. gondii* has among the smallest mitochondrial genome contents, encoding only three highly hydrophobic proteins[12,16], we reasoned that any such exit tunnel adaptations would be especially prominent in our structure. We thus carefully analysed the full exit tunnel while drawing a comparison to the mitoribosome structure of *Tetrahymena thermophila*[29], a ciliate with a large mitochondrial genome that encodes both soluble and membrane

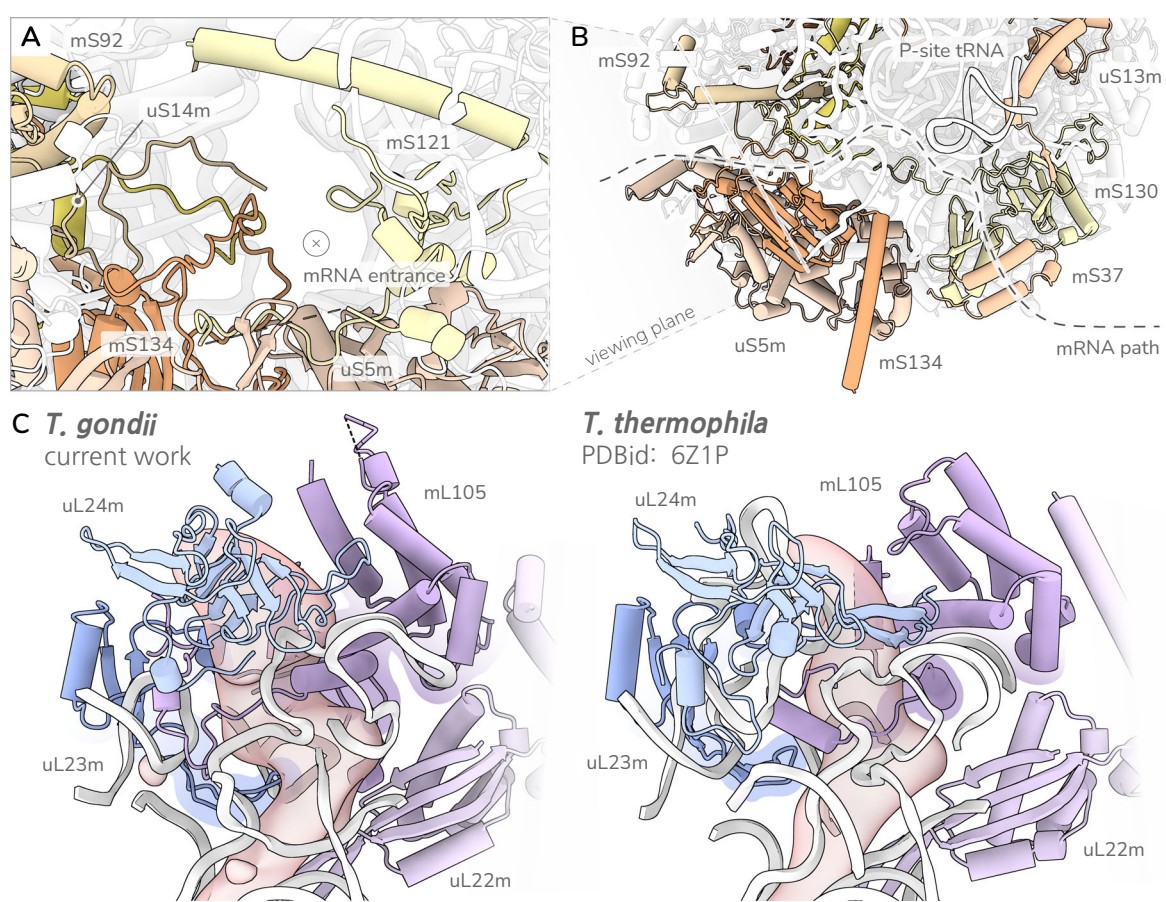

**Fig. 5 | Remodelling of mRNA channel and exit tunnel. A** View of mRNA protein entrance populated primarily by extensions from uS14m, mS92 and mS134 alongside the conserved uS5m interacting with mS121. **B** Lateral view of mRNA binding path, with entrance to the left, exit to the right, highlighting presence of P-site tRNA modelled from the density. mS130 containing tandem ApiAP2-like TF fold in the position of lost uS7m by the mRNA exit. All proteins shown as cartoons and coloured individually by chain. **C** Comparison of ribosomal exit tunnel (lower) and vestibular area (upper) in comparison to the closest structural reference *T. thermophila*. rRNA shown as white cartoon. Solvent accessible surfaces within 8 Å of displayed protein coloured red. Vestibular area conserved with minor rRNA modifications. *T. gondii* late exit tunnel featuring a prominent kink prohibiting helical formation. All proteins shown as cartoons and coloured individually by chain.

proteins[38], and which is a close relative of the Myzozoa lineage. We observed broad similarities with the vestibular area of the *Tetrahymena* mitoribosome (Fig. 5C), with the exception of a five-residue loop insertion in uL23m *in T. gondii* (Supplementary Fig. 3). This insertion is primarily conserved among Apicomplexa, thus most likely was acquired following their divergence from other Myzozoa. The insertion, coupled with a stronger helical propensity in the anchoring helix of mL105, produces an additional kink in the late exit tunnel. Such a kink inevitably prohibits helical formation in the nascent chain until the vestibular area. It is not clear, however, how the inhibition of helix formation late in the tunnel would aid the process of co-translational protein insertion. Further analysis of chemical properties revealed that the late *T. gondii* tunnel and vestibular area is less negatively charged than *Tetrahymena* (Supplementary Fig. 13). This follows from the rRNA reduction as the negatively charged phosphate backbone is replaced by proteins. Likewise, the replacement of rRNA with proteins affects the distribution of hydrophobic patches across the tunnel, however there is no clear pattern that distinguishes the two structures (Supplementary Fig. 13). Rather, we claim that the generally conserved vestibular area, coupled with only minor changes to the late exit tunnel, across these two species with widely different mitochondrial genome contents, argues against any specific trend of tunnel adaptation.

## Discussion

Our study addresses how the apicomplexan mitoribosome forms and function despite the disruptive rRNA fragmentation imposed by its reduced and scrambled mt-genome and reveals adaptation mechanisms for accommodating ribosomal rRNA molecules that are too small for independent association. These mechanisms are likely conserved in other myzozoans. We observe several adaptations promoting rRNA molecule association, either directly via protein-rRNA interactions, or mediated via the surprising case of structurally incorporated rRNA poly-A tails acting as handles. Interestingly, since the poly-A tails are not encoded in the mt-genome but still form an integral part of the mitoribosome, polyadenylation of rRNA is revealed as an effective trait that supports mt-genome reduction. Furthermore, by identifying the 53 rRNA molecules and aligning them against the *Toxoplasma* mt-genome blocks, we revealed the 're-use' of rRNA sequences as another mechanism supporting mt-genome reduction, and a testament to the malleability of rRNA and the remarkable structural plasticity of mitoribosomes. For nine of the 53 rRNA we could not identify a corresponding sequence in the mt-genome, leaving an open question regarding their source. While identified both in the RNA sequencing and independently in the density, it cannot be fully excluded that those nine might be preparation artefact.

Further mitigation for the short rRNA molecules is provided through the proliferation of protein extensions, a hallmark of mitoribosomal evolution, although previously only discussed in the context of protein recruitment to the ribosome. The protein extensions described here instead play a fundamental role in rRNA stabilisation and are therefore observed as a direct driver of mitoribosome divergence. Protein extensions also play a pivotal role in the remodelled subunit interface which, when coupled to the recruitment of proteins with transcription factor-like folds, drive a shift from rRNA to protein mediated subunit interactions.

A curious case is the *T. gondii* mitoribosome is the incorporation of four proteins, mL161, mS130, mL143 and mL153 (Fig. 4C, Fig. 5B, Supplementary Data 2, Supplementary Fig. 14) containing typical apicomplexan AP2 transcription factor-like domains (ApiAP2[33]). The family of proteins containing ApiAP2 domains is expanded in Apicomplexa and related organisms[33,39] and its members are critical for transcriptional switching during stage transition within the complex life cycle of apicomplexans. For example, ApiAP2 members mediate *Plasmodium* differentiation to the transmissible mosquito stages[40],

and *Toxoplasma* conversion into its persistent stages[40]. Thus far all ApiAP2 members were considered to be transcription factors that bind DNA[39,40] and here we present a distinct case of members that were recruited to perform a different function, suggesting a more diverse role for this family in Apicomplexa and possibly in other organisms, than currently believed. Interestingly, the interaction of these four mitoribosomal ApiAP2s with nucleic acid, rRNA in this case, is not via the canonical ApiAP2 DNA-binding features, rather they are primarily used as scaffolds from which extensions have grown to mediate rRNA interactions. Further, it is unlikely that these proteins simultaneously serve as transcription factors, as they lack other functional domains such as the AT hook, zinc-finger, ACDC often found in AP2 transcription factors[34], and further possess long extensions likely to cause misfolding or aggregation if outside the mitoribosome context. Given the expansion of AP2 domain-containing proteins in apicomplexans and across Chromalveolata[39], it is intriguing to consider if other members might mediate RNA regulation also outside of the mitochondrial compartment.

An additional apicomplexan research horizon opened by our structure is the potential of developing apicomplexan specific mitoribosome inhibitors as leads for drugs. *T. gondii*, is an important pathogen, and is closely related to other medically important apicomplexan parasites, such as the malaria causing *Plasmodium* spp, with which most of the herein characterised proteins and domains are conserved. Inhibition of organellar ribosomes is already a promising avenue for anti-apicomplexan drug development[41]. For example, it was already proposed based on structural predictions that differences between human and parasite mitoribosome will be responsible for different sensitivity to drug[5]. It is thus tempting to suggest that the divergent nature of the apicomplexan mitoribosome that we expose here might present exciting opportunities for drug development.

## Methods

### Parasite cell culture

*Toxoplasma gondii* tachyzoites were grown at 37 °C with 5% $CO_2$ in either Human foreskin fibroblast (HFFs) or Vero cells in DMEM (Dulbecco's Modified Eagle Medium) containing 4.5 g/L glucose, supplemented with 10% FBS (Foetal Bovine Serum) (v/v), 4mM L-glutamine and penicillin/streptomycin as antibiotics. Anhydrotetracycline (ATc) was added into the growth medium to a final concentration of 0.5 μM when indicated.

### Transgenic cell lines

A bL12m tagged line - CRISPR-Cas9 was employed to insert a C-terminal 3x FLAG epitope tag to the TGGT1_251950 locus (bL12m), guide RNA targeting the annotated stop codon was identified using the online ChopChop tool (https://chopchop.cbu.uib.no/) and cloned into the Cas9-YFP expressing plasmid with a U6 promoter (Tub-Cas9YFP-pU6-ccdB-tracrRNA)[42]. The 3xFLAG epitope along with CAT (chloramphenicol acetyltransferase) repair cassette was amplified by PCR from a pLIC-TEV-3xFLAG-CAT plasmid[2]. *T. gondii* RH based line TATiΔ*ku80*[43] was transfected with the guide RNA plasmid and a PCR product mixture. Integrants were selected with Chloramphenicol and cloned using serial dilution. Single clones were tested for genetic modification at the endogenous locus through PCR analysis using locus specific primers (Supplementary Fig. 1, Supplementary Data 6).

Knockdown cell lines - CRISPR-Cas9 based promoter replacement strategy was employed similar to above, with the changes that the guide RNA targeted the start codon and repair cassette was a PCR product amplified from pDT7S4[43] containing the ATc repressible T7S4 promoter and a DHFR selectable cassette. The above bL12m-3xFLAG cell line was used as parental to generate both knockdown lines. Positive transfectants were selected using Pyrimethamine and cloned and confirmed as above (Supplementary Fig. 8). All primers used for creating transgenic cell lines are listed in Supplementary Data 6.

## Immunoprecipitation

Freshly egressing parasites grown in Vero cells were harvested and incubated for 2 h in a lysis buffer (20 mM HEPES-KOH pH 7.5, 200 mM NaCl, 30 mM MgCl$_2$, 2% β-DDM) followed by centrifugation at 16,000 × g at 4 °C for 30 min. The lysate (supernatant) was incubated with washed anti-FLAG M2 affinity agarose beads (Merck) overnight at 4 °C with rotation. The unbound fraction was discarded, and the affinity beads were washed thrice with wash buffer (20 mM HEPES-KOH pH 7.5, 200 mM NaCl, 30 mM MgCl$_2$, 0.05% β-DDM) followed by elution with FLAG peptide (1 mg/ml). The obtained elute was concentrated to ~20 μL using a vivaspin500 filter (100-kDa MWCO). Samples were vitrified by plunge freezing into liquid ethane using a Vitrobot and Quantifoil R2/2 grids with 2-nm carbon support layer after 3-s blotting.

## Electron cryo-microscopy and data processing

CryoEM data collection was performed on a Titan Krios operated at 300 kV at 0.83 Å/pixel and a total exposure of 35 electrons/Å$^2$, resulting in 30,431 movies. (40 frames each). CryoEM data processing was performed in cryoSPARC v4.4 (Supplementary Fig. 2), with initial references generated in cryoSPARC Live. Following motion-correction and contrast transfer function estimation, blob picking was performed on a subset on 12,250 micrographs and 699,000 particles were extracted and subjected to 2D classification for initial reference generation. Nine ribosome-containing references (out of 50) were selected to allow reference-based particle picking on the full dataset, yielding 2,219,818 particles. Following 2D classification (Fourier-cropped to 128 pixels), 375,745 particles were selected from 7 (of 50) classes, separating ribosomes from junk and micelle particles, and extracted unbinned (700 pixel box). High resolution structures were obtained by masked, non-uniform refinements of the LSU, SSU and two SSU sub-areas, with resulting maps ranging in resolution from 2.2 to 2.7 Å. We note that due to anisotropy, the resolution of the maps in real space appears lower than suggested by the nominal values. The resulting maps displayed anisotropic resolution due to preferred orientation and were modified using deepEMhancer[44], yielding maps with improved interpretability for model building.

## Model building and refinement

To model the fragmented rRNA, the mitoribosomal rRNA from *Tetrahymena thermophila*[29], PDBID: 6Z1P was manually fitted into the density and cropped to size. Fragment sequences were assigned following a build-and-blast approach. Briefly, an initial 'best guess' sequence is assigned based on density fit, and searched against the *T. gondii* assembly reads available under SRA accession SRR9200762, using BLAST[45]. Fragment termini and poly-A tails were initially assessed from the density alone and then verified using RNA seq (Supplementary Data 4). Nucleic acid bound in the SSU head, close to the mS135/136 heterodimer, is modelled as RNA with sequence of best fit, but density could also be interpreted as DNA. Final sequences were interactively relaxed into density using Coot and subsequently refined as outlined below.

In the cases where we observed A to G discrepancies between the genomic sequence reads and our density, we assessed the abundance of such discrepancies in our rRNA sequencing reads. To account for sequencing error, we constructed an error substitution matrix by calculating a substitution matrix from manually curated read alignments. Assessing 250.000 independent and aligned sites from our data we estimate that G to A errors occurred at a frequency of 9e-4. Based on this null-frequency of errors we assessed the probability of observing the given amount of G to A transitions for the suspected discrepancies by modelling all such observed substitutions as a Bernoulli trial. P-values and observed densities for all probable discrepancies are listed in (Supplementary Fig. 2A, Supplementary Data 4).

Proteins were modelled by initially assigning protein fragments to the density using ModelAngelo[46], followed by a query of the de-novo modelled sequences against annotated proteins acquired from ToxoDB [https://toxodb.org/toxo/app]. Initial models for likely sequence hits were downloaded from the AlphaFold Protein Structure Database[47,48] and fitted against the de-novo modelled chain. The fitted models were then manually trimmed and edited in Coot[49] to optimise model geometry and fit to density. This process was then iterated, excluding modelled regions from further rounds of ModelAngelo prediction. Once no new density could be assigned this way a build-and-BLAST approach was carried out for the final chain assignments in which residue identities were initially assigned manually. For proteins with no annotated open reading frame the initial modelled sequence was instead searched against the TGGT1 genome using tblastn implemented in the ToxoDB website. Final regions of protein density which could not be assigned with a known sequence were modelled as poly-alanine. Initial model refinement was performed using ISOLDE in ChimeraX[50]. Subsequent manual model refinement was performed in Coot[49], followed by iterative real-space refinement in PHENIX using secondary structure restraints[51]. Q-Scores were calculated in Chimera[13] using MapQ[14]. Models were refined against sharpened maps (CryoSPARC). For the LSU, the three maps originating from masked refinements were combined in ChimeraX and refinement was performed against the composite map to generate a single SSU model.

## RNA extraction and cDNA library preparation

Elute from immunoprecipitated sample as described above was resuspended in TRIzol™ Reagent (Invitrogen) and the RNA was isolated using the acid-guanidium-phenol based method[52]. 200 ng of RNA were used to generate sequencing library using the NEBNext® 600 Multiplex Small RNA Library Prep Set for Illumina® Kit (New England BioLabs) according to the manufacturer's protocol. Sequencing was performed by Glasgow Polyomics on a Illumina NextSeq2000 instrument with 100 bp read length for each pair.

## Analysis and quantification of rRNA fragments by RNA sequencing

Reads in fastq format were trimmed with cutadapt[53] to remove adaptors ligated to 3'end by searching the sequences AGATCGGAAGAGC (for first-in-pair reads) and GATCGTCGGACT (for second-in-pair reads). Next, read pairs were merged using the command merge-pairs from the vsearch package[54]. The merged reads were then aligned to the sequences inferred from CryoEM using bowtie2[55] in 'local' alignment mode and with alignment parameters adjusted to deal with short reference sequences. Finally, plots and summaries of reads aligned per library and per rRNA reference were produced with custom scripts in R. This pipeline was implemented in a mamba (conda) environment as a snakemake workflow[56] available at https://github.com/glaParaBio/cryoem-mitoribosome-rnaseq together with the exact programme versions. The DOI for the github repository is, https://doi.org/10.5281/zenodo.14202883 [57].

## qRT-PCR

Parasites grown without or with ATc (2 days), were filtered through a polycarbonate 3 μM filter and collected through centrifugation. RNA was extracted using the RNeasy Mini kit (Qiagen) with DNase I treatment (Thermo Fisher). cDNA synthesis was done using Applied Biosystems™ High-Capacity RNA-to-cDNA™ Kit followed by qPCR reaction set up using Applied Biosystems™ Power SYBR™ Green PCR Master Mix. Gene specific primers were designed and used for the qPCR reactions. Relative expression of the genes of interest in ATc untreated versus treated samples was calculated using the ΔΔCt method with actin mRNA used as internal control. Three independent replicates were performed for each experiment. Data was plotted using Graphpad Prism 9.4.1. Primers used listed in Supplementary Data 6.

## Native & SDS-PAGE and Western Blot

For Blue-Native PAGE, parasites were solubilised in 1% β-DDM (w/v) in 750 mM aminocaproic acid solution on ice for 10 min. The resulting lysate was centrifuged at 16,000 × g at 4 °C for 30 min. Cleared lysate was mixed with Coomassie G250 to a final concentration of 0.25%.

For Clear-Native PAGE, freshly harvested parasites were resuspended in solubilisation buffer composed of 50 mM NaCl, 2 mM 6-aminohexanoic acid, 50 mM imidazole, 2% (w/v) βDDM, 1 mM EDTA−HCl pH 7.0 and allowed to incubate for 10 min on ice. Samples were then centrifuged at 16,000 × g in a table-top centrifuge for 15 min at 4 °C. Cleared lysates were transferred to a fresh tube and mixed with loading dye containing glycerol and Ponceau S (final concentration of 6.25% and 0.125% respectively).

Samples were loaded onto a precast 4–16% or 3–12% Bis-Tris Polyacrylamide Native gel (NativePAGE™, Bis-Tris, 1.0 mm, Mini Protein Gels). The gel was run at 80 V for 1 h followed by 250 V for 2 h. Proteins were electroblotted onto a charged PVDF membrane (0.45 μm, Hybond, Merck, Gillingham, EN, UK) using wet transfer method (100 V; 60 min) in Towbin buffer (0.025 M Tris 0.192 M Glycine; 10% Methanol).

Blots were then blocked with 5% skim milk in PBS 0.1% Tween 20 solution and probed with desired antibodies: anti-FLAG (1:2000, mouse monoclonal ANTI-FLAG® M2, F3165, Sigma-Aldrich); 2° anti-rabbit horseradish peroxidase (HRP) conjugated antibody (anti-rabbit (1:10000, W4011, Promega) and anti-mouse (1:10000, W4021, Promega) and developed with Pierce ECL Western Blotting Substrate (Thermo Scientific, Paisley, UK).

For SDS-PAGE, parasites were resuspended in Laemmli sample buffer containing β-mercaptoethanol and loaded onto a 12% SDS-polyacrylamide gel. Proteins were blotted onto a nitrocellulose membrane, blocked with 5% skim milk and probed with desired primary (anti-FLAG (1:5000, mouse monoclonal ANTI-FLAG® M2, F3165, Sigma-Aldrich); anti-TgTom40[58] (1:2000); anti-TgCDPK1[59] (1:2000) and secondary antibodies (IRDye 800CW and IRDye 680RD (1:10,000, LIC-COR, Lincoln, NE, USA) and detected with Odyssey CLx imaging system.

## Fluorescent immunostaining and microscopy

Confluent HFFs on glass coverslips were infected with parasites and fixed after 1 day with 4% paraformaldehyde. Fixed cells were permeabilised and blocked with 0.2% Triton-X100 in PBS containing 2% BSA (Bovine Serum Albumin). The coverslips were then incubated in primary antibody cocktail (Monoclonal Anti-FLAG® M2 (1:5000; F3165, Sigma-Aldrich); anti-TgTom40[58] (1:2000), washed 3X with 0.2% Triton-X100 in PBS, followed by incubation in secondary antibody cocktail (Alexa Fluor Goat anti-Mouse 594 Invitrogen (1:1000) and Alexa Fluor Goat anti-Rabbit 488 Invitrogen (1:1000)). Coverslips were mounted on glass slides using DAPI Fluoromount-G® mounting solution (Southern Biotech, Cat. nr. 0100–20). Images were taken via a DeltaVision Core microscope with a 100X objective and z-stacking. Images were deconvolved using SoftWoRx software, processed using Fiji software[60].

## Plaque assays

HFF monolayer on six cm dishes were infected with ~200 parasites and allowed to grow continuously for 10 days with minimal physical disturbance, either with or without ATc (0.5 μM). Cells were then washed with PBS and fixed with chilled methanol followed by staining with 0.4% crystal violet solution. Pictures were taken against a bright white light background using iphone14 phone camera.

## Replication assays

For growth analysis via measuring parasite replication, respective cell lines were grown in the absence or presence of ATc for 48 h followed by using these parasites to re-infect a confluent HFF monolayer on a coverslip for 24 h in same conditions. Infected cells were then washed with PBS to remove extracellular parasites, and fixed with 4% PFA.

Parasite vacuoles on the fixed coverslips were visualised by immunostaining and fluorescent microscopy as described above using anti-GAP45 antibody. Vacuoles containing 2, 4 or 8+ parasites were manually counted for a total number of 100 vacuoles for each condition. Three independent biological experiments were performed.

## Mitochondrial translation assay

As described[2,3] this assay compares between the mitochondrial translation dependent complex IV assembly and activity and the mitochondrial translation independent assembly of Complex V we use native migration and in gel activity assay or antibody detection respectively. Briefly, for in-gel complex IV activity native-PAGE was run as above. NativeMark™ Unstained Protein Standard (Invitrogen) was used as a molecular weight ladder. The gel was incubated in a freshly prepared solution containing 50 mM $KH_2PO_4$, pH 7.2, 1 mg/ml cytochrome c, 0.1% (w/v) 3,3′-diaminobenzidine tetrahydrochloride (DAB), which was pre-warmed to 37 °C, until a brown precipitate is visible (roughly at the size of Native Complex IV). For Complex V assembly we followed the native-PAGE and western blot protocol above and used anti-ATPβ (1:2000, rabbit, Agrisera AS05 085) to detect complex V.

## Complex II assay

Complex II activity was analysed using a spectrophotometric assay as recently described[61]. Parasites were collected after filtering through a 3 μM polycarbonate filter and counted using a Neubauer chamber. Equal number of parasites were resuspended in 100 μL lysis buffer (70 mM sucrose, 220 mM mannitol, 10 mM $KH_2PO_4$, 5 mM $MgCl_2$, 2 mM HEPES, 1 mM EGTA, 0.2% w/v bovine serum albumin (BSA) and 0.2% w/v digitonin, pH 7.2) and incubated for 1 h on a rotator at 4 °C. Lysed samples were then mixed with the assay buffer (20 mM succinate, 25 mM $KH2PO_4$, 0.002175% w/v DCPIP, 1 μM atovaquone, 2.5 mg/mL BSA and 40 μM decylubiquinone) to make up to 1 mL final volume. For each sample absorbance was recorded for 40 min with 2 min interval. Enzymatic activity was calculated using DCPIP (2,6-Dichlorophenolindophenol) extinction coefficient of 19.1 mM$^{-1}$ cm$^{-1}$.

## Mass spectrometry analysis

Anti-FLAG agarose beads bound to bL12m-FLAG tagged protein were washed several times with ammonium bicarbonate to remove excess DDM in the sample followed by on-bead trypsin (Promega) digestion. A portion of the digested peptides were analysed by LC-MS. LC-MS analysis was performed by the Mass Spectrometry and Proteomics Facility (University of St Andrews). A portion of anti-FLAG agarose beads with no bound bL12m-FLAG was used for immunoprecipitation control (n = 1) and was sent for mass-spectrometry analysis too. Analysis of peptide readout was performed on a Fusion Lumos Mass Spectrometer (Thermo Scientific) with FAIMS coupled to an Ultimate 3000 Nano RSLC system (Thermo Scientific). LC buffers used are the following: buffer A (0.1% formic acid in Milli-Q water (v/v)) and buffer B (80% acetonitrile and 0.1% formic acid in Milli-Q water (v/v)). Reconstituted samples were loaded at 10 μL/min onto a trap column (100 μm × 2 cm, PepMap nanoViper C18 column, 5 μm, 100 Å, Thermo Scientific) which was equilibrated with 0.05% Trifluoroacetic acid. The trap column was washed for 3 min at the same flow rate and then the trap column was switched in-line with a Thermo Scientific, resolving C18 column (75 μm × 15 cm, PepMap RSLC C18 column, 2 μm, 100 Å). Peptides were eluted from the column at a constant flow rate of 300 nl/min using a linear solvent gradient using the following gradient: linear 2–40% of buffer B over 40 min, linear 40–95% of buffer B for 4 min, isocratic 95% of buffer B for 6 min, sharp decrease to 2% buffer B within 0.1 min and isocratic 2% buffer B for 10 min. The FAIMS interface was set to −40 V and −65 V at standard resolution. The mass spectrometer was operated in DDA positive ion mode with a cycle time of 1.5 s. The Orbitrap was selected as the MS1 detector at a resolution of 120,000 with a scan range of from m/z 375 to 1500. The RF-lens was set to 30%. Peptides with charge states 2–5 at a minimum intensity

threshold of 5e3 were selected for MS/MS fragmentation. Once selected, a dynamic exclusion for 45 s with a 10 ppm mass tolerance was applied. The Iontrap was selected for data dependent MS2 fragmentation with an isolation window of m/z 0.7 and no isolation offset. The peptides were fragmented with fixed HCD as activation type at 28% HCD Collision Energy.

The data was searched using Mascot against the ToxoDB database (ToxoDB Version-64, *T. gondii* GT1 strain annotated proteins database) with Carbamidomethylation on Cysteines as fixed modification and Oxidation on Methionines as variable modification. The mass tolerance was set to 20 ppm for the MS scan and to 0.6 Da for the MS/MS scan. The mass spectrometry proteomics data have been deposited to the ProteomeXchange Consortium via the PRIDE partner repository with the dataset identifier PXD053944.

### Reporting summary
Further information on research design is available in the Nature Portfolio Reporting Summary linked to this article.

## Data availability
The atomic coordinates generated in this study were deposited in the Protein Data Bank (PDB) under accession numbers 9I05 (LSU) and 9HQV (SSU). The cryo-EM maps generated in this study have been deposited in the Electron Microscopy Data Bank (EMDB) under the accession numbers EMD-52551 (LSU) and EMD-52348 (SSU). The sequencing data generated in this study is available on the ENA archive with the accession code PRJEB72258. The mass spectrometry proteomics data generated in this study have been deposited to the ProteomeXchange Consortium via PRIDE with the dataset identifier PXD053944. Source data are provided as a Source Data file. Full versions of all gels are provided in the source file. Source data are provided with this paper.

## Code availability
The code for the analysis of the sequencing data is available from the github repository at https://github.com/glaParaBio/cryoem-mitoribosome-rnaseq and as Zenodo entry 14202883 [57].

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

## Acknowledgements

We thank EuPathDB for free access to well annotated genomes. We thank Katarzyna Modrzynska for advice while generating the small rRNA library. We thank Glasgow Polyomics for performing the sequencing and Mass Spectrometry and Proteomics Facility (University of St Andrews) for LC-MS experiment. We thank the Swedish National cryoEM facility at SciLifeLab (funded by the KAW, EPS and Kempe foundations), for instrument access and assistance with data acquisition. The Wellcome Centre for Integrative Parasitology is supported by core funding from the Wellcome Trust [104111]. This work was supported by Wellcome Investigator Award (217173_Z_19_Z) (to L.S.); by a FutureScope Fellowship by the Wellcome Centre for Integrative Parasitology (to A.M.); and S.S. was supported by Swiss National Science Foundation Early Postdoc Mobility Fellowship (Grant nr. 200158).

## Author contributions

L.S. conceived the idea; L.S., A.M., S.S. designed the experiments; S.S., A.M. performed cell culturing and sample isolation; S.S., M.F.S., J.O. performed genetic manipulation of parasites and cell biological and biochemical phenotypic assays; A.M., S.S. performed cryoEM sample preparation; A.M. performed data collection, data processing; A.M., S.S., V.T. performed atomic model building and analysis; A.M. performed atomic model refinement; V.T., S.S. performed data analysis and generated structure figures; L.S. and V.T. wrote the manuscript with contributions from S.S. and A.M.; D.B. analysed RNA Sequencing data. All authors contributed to figure and table preparation and to the writing of methods.

## Competing interests
