## [Transparent Peer Review file · Nature Communications]

Numerous rRNA molecules form the Apicomplexan mitoribosome via repurposed protein and RNA elements

Corresponding Author: Professor Lilach Sheiner

This manuscript has been previously reviewed at another journal. This document only contains information relating to versions considered at Nature Communications. Mentions of the other journal have been redacted.

Version 0:

Reviewer comments:

Reviewer #1

(Remarks to the Author)

In this manuscript, Shikha et al. has solved the structure of the mitoribosome from parasitic *T. gondii* and modeled proteins and rRNA into its density map. This result led to the discovery of novel features of *T. gondii* mitoribosome including fragmentation of rRNA into small fragments (strikingly, 53 fragments of varying lengths), rRNA expansion and post-transcriptional modifications of rRNA through poly-A tail addition which differ from bacterial ancestor. The author also reported the elaborate protein-protein and protein-rRNA interactions within the mitoribosome. The *T. gondii* mitoribosome also has a reduced form through reduction of proteins and compensation of proteins that possess transcriptional factor-like fold. Overall, these results highlight the novelty and discrepancies of *T. gondii* mitoribosome from other apicomplexans. The results are clearly presented. I only have minor comments and suggestions before I can support its publication.

For data processing, has the author performed 3D classification to further separate junk or subclasses from the dataset? If so, it would be appropriate to mention in the method section and also include in Fig ED2.

For rRNA modeling, the author mentioned that the rRNA fragments were manually fitted and modeled based on *T. thermophila* structure. Has the author performed real-space refinement for the rRNA models using software such as ERRASER2? Author could also score how the rRNA model fits into the density map using software such as MapQ. This will provide confidence to the reader for your rRNA modeling.

It would be informative to show the local resolution for the rRNA core.

Line 76-77 “*T. gondii* mitoribosomes were purified through immuno-precipitation via the endogenously FLAG-tagged bL12m”.

It would be nice to define what bL12m stands for in the text for general readers.

Line 83-83 “Among the proteins 55 are clade-specific (Table S3)”,
Should it be “Among the proteins, 55 are clade-specific”?

Line 139-140 “also observed seven rRNA expansion segments, three in the LSU and four in the SSU”.
I see that there are 3 and 5 highlighted as yellow for LSU and SSU, respectively. Is it a typo here?

Line 280 “a strand of nucleic acid (Fig 4B)”.

Author mentioned that this could either be RNA or ssDNA. I see that the author fitted an atomic (or possibly pseudo-atomic) model into a density map here. Is it just a backbone being modeled into a density map? I think it should be mentioned for clarity.

Line 326-327 “The missing uS3m binding site is occupied by several terminal extensions from surrounding proteins (uS14m,

mS92 and mS134),”

Do the terminal extensions here have sequences similar to missing uS3m? Do the authors see the density for these terminal extensions in the cryo-EM map? A supplement figure showing the termini fitted in density will be helpful.

Reviewer #2

(Remarks to the Author)

In this revised version of the manuscript, the authors have responded satisfactorily to all my previous queries and, in my opinion, also those of the other reviewers. Therefore, I recommend this article to be accepted for publication in Nat Comm.

Reviewer #3

(Remarks to the Author)

The authors have satisfactorily addressed all comments and conducted additional biochemical analyses that support their claims. I have no further major comments and endorse the publication of the manuscript.

Minor issues:

Please review the references throughout the text. For instance, missing reference in line 1022.

Reviewer #4

(Remarks to the Author)

In the manuscript "A mitoribosome with 53 rRNA molecules requires extensive protein and RNA repurposing to function" by Shikha et al., the authors present a high-resolution structure of the *Toxoplasma gondii* mitoribosome using cryoEM and RNA sequencing technologies. They determined the 3D structures of all 53 rRNAs, revealing that the rRNAs in the mitoribosome core possess poly-A tails. The enzyme responsible for this polyadenylation was identified as TgmtPAP1. The study highlights the significance of rRNA polyadenylation, protein extensions, and proteins with transcription factor-like domains in the formation and stability of the mitoribosome. This work represents the first successful resolution of the *T. gondii* mitoribosome structure, marking a significant advancement in the field of *T. gondii* research.

Major Points:

1. Please include quantification for both plaque size and number in all plaque assays.

2. Please add quantification of all western blots.

It seems that the bands for Complex V in Figure 1E appear overexposed. Please verify that no overexposed blots are presented, as this can lead to inaccurate quantifications.

For the TgmtPAP1 KD in Figure 1E and mL161 KD in Figure ED11C, it appears that normalizing the bands for Complex V to the bands for TgTom40 may show a decreased level of Complex V after the knockdowns. Please include quantification to verify this observation. If there is a decrease, please discuss how this might affect Complex V, given that the *T. gondii* mitoribosome is not reported to translate proteins for Complex V.

3. Can the author specifically state in the text which three proteins involved in the electron transport chain are translated by that *T. gondii* mitoribosome? Are they COX1, COX3 and COB? COX1 and COX3 are components of Complex IV, and COB is part of Complex III. When TgmtPAP1 or other mitoribosome proteins are knocked down, the authors demonstrated that the assembly and activity of complex IV are disrupted using functional assays, while Complex II activity, indicated by SDH activity assay, remains unaffected. Could the authors include an assay to evaluate the assembly and activity of complex III?

4. How can you ensure that the rRNA fragmentation observed in the cryoEM structure and RNA sequencing data is not an artifact of the isolation process? Did you use any controls to rule this out?

5. For the mass spectrometry analysis, the data was searched only against the ToxoDB database. Since the parasites were grown in Vero cells (host cells), there could be contamination from host proteins. If the mitoribosome proteins from Vero cells are not drastically different from those of the parasites, their electron density in the cryoEM analysis could be indistinguishable. Please analyze the mass spectrometry data against a protein database for Vero cells to determine if there is any contamination from the host cells.

Minor points:

1. Figure 1: when labeling mRNA entrance and tunnel exit, please use a line or arrow to indicate their locations, so that the labels and their backgrounds do not obscure the structure.

2. Line 255: It is not accurate to state that "Several transcription factors are...." As the author stated in the discussion, "it is unlikely that these proteins simultaneously serve as transcription factors, as they lack other functional domains such as the AT hook, zinc-finger, ACDC often found in AP2 transcription factors." It would be more accurate to say, "Proteins contain transcription factor-like domains are...."

3. Lines 264 and 265: Please add an explanation why mL151 has a BolA-like domain, and mL161 has an ApiAP2-like domain. Do mL151 and mL161 have DNA-binding-like domains or domains that mediate transcription?

4. Line 410: In this paragraph, the authors discuss targeting the *T. gondii* mitoribosome for drug development. The paper compares the differences between *T. gondii* mitoribosome and bacterial mitoribosome but does not mention differences between *T. gondii* and human mitoribosomes. Please consider adding a description of why the *T. gondii* mitoribosome might be different enough from the human mitoribosome to serve as drug targets.

Reviewer #5

(Remarks to the Author)

Version 1:

Reviewer comments:

Reviewer #1

(Remarks to the Author)

The authors have addressed all my critiques. Therefore I support it's publication.

Reviewer #4

(Remarks to the Author)

The authors have thoroughly addressed all of my concerns. I have no further suggestions, and I recommend the paper for publication. This is an excellent paper. Congratulations to the authors!

Reviewer #5

(Remarks to the Author)

made.

Please find our response to the reviewer comments. We answer each comment below in blue/bold text under each comment. The descriptions of any new data added, and any text/figure/table revised are underlined, line numbers (corresponding to the version that contains track-changes) are provided for text revisions.

REVIEWER COMMENTS

Reviewer #1 (Remarks to the Author):

In this manuscript, Shikha et al. has solved the structure of the mitoribosome from parasitic *T. gondii* and modeled proteins and rRNA into its density map. This result led to the discovery of novel features of *T. gondii* mitoribosome including fragmentation of rRNA into small fragments (strikingly, 53 fragments of varying lengths), rRNA expansion and post-transcriptional modifications of rRNA through poly-A tail addition which differ from bacterial ancestor. The author also reported the elaborate protein-protein and protein-rRNA interactions within the mitoribosome. The *T. gondii* mitoribosome also has a reduced form through reduction of proteins and compensation of proteins that possess transcriptional factor-like fold. Overall, these results highlight the novelty and discrepancies of *T. gondii* mitoribosome from other apicomplexans. The results are clearly presented. I only have minor comments and suggestions before I can support its publication.

For data processing, has the author performed 3D classification to further separate junk or subclasses from the dataset? If so, It would be appropriate to mention in the method section and also include in Fig ED2.

Neither 3D classification nor heterogeneous refinement resulted in a further separation of heterogeneity, whereas further partitioning of the 2D classified particles was detrimental to the final resolution and map quality. We mention this in the revised description of Fig ED2 as the reviewer proposed.

For rRNA modelling, the author mentioned that the rRNA fragments were manually fitted and modelled based on *T. thermophila* structure. Has the author performed real-space refinement for the rRNA models using software such as ERRASER2? Author could also score how the rRNA model fits into the density map using software such as MapQ. This will provide confidence to the reader for your rRNA modelling.

As requested, we scored the rRNA models using MapQ, which reported a Q-Score of 0.64 for the LSU (62,8525 atoms; sigma value: 0.6) and 0.54 for the SSU (35,437 atoms), which are typical for RNA cryo-EM map features at around 2.5 and 3 Å, respectively (PMID: 32042190). We added these values to the revised Table S2 and describe the software usage in the methods (678).

It would be informative to show the local resolution for the rRNA core.

We now display this as requested as an added panel H to Fig.ED2.

Line 76-77 “T. gondii mitoribosomes were purified through immuno-precipitation via the endogenously FLAG-tagged bL12m“. It would be nice to define what bL12m stands for in the text for general readers.

We have added to the main text an explanation that this is a large ribosomal subunit protein (line 78).

Line 83-83 “Among the proteins 55 are clade-specific (Table S3)”,
Should it be “Among the proteins, 55 are clade-specific”?

Corrected.

Line 139-140 “ also observed seven rRNA expansion segments, three in the LSU and four in the SSU”.

I see that there are 3 and 5 highlighted as yellow for LSU and SSU, respectively. Is it a typo here?

Thank you for pointing this out, the additional sequence highlighted in yellow in the SSU was the sequence for which we cannot exclude it being DNA rather than RNA (mentioned in the next comment too). We thus removed it from the revised Fig ED6 (that aims to only show RNA).

Line 280 “a strand of nucleic acid (Fig 4B)”.

Author mentioned that this could either be RNA or ssDNA. I see that the author fitted an atomic (or possibly pseudo-atomic) model into a density map here. Is it just a backbone being modelled into a density map? I think it should be mentioned for clarity.

We now clarify that this nucleic acid is modelled as RNA with sequence of best fit, but that the density could also be interpreted as DNA in the revised method section under “Model building and refinement” (line 651).

Line 326-327 “The missing uS3m binding site is occupied by several terminal extensions from surrounding proteins (uS14m, mS92 and mS134),”. Do the terminal extensions here have sequences similar to missing uS3m? Do the authors see the density for these terminal extensions in the cryo-EM map? A supplement figure showing the termini fitted in density will be helpful.

We clarified in the revised text that there is no sequence similarity (line 352), and we added panel D to Fig ED8 showing the termini fitted in density following the reviewer’s request.

Reviewer #2 (Remarks to the Author):

In this revised version of the manuscript, the authors have responded satisfactorily to all my previous queries **and, in my opinion, also those of the other reviewers**. Therefore, I recommend this article to be accepted for publication in Nat Comm.

We wanted to highlight that this reviewer is not only satisfied with our originally revised manuscript, but also supports that our first round of revision already addressed all the reviewers’ comment provided upon consideration of the publication in [redacted], including the reviewer that needed to be replaced.

Reviewer #3 (Remarks to the Author):

The authors have satisfactorily addressed all comments and conducted additional biochemical analyses that support their claims. I have no further major comments and endorse the publication of the manuscript.

Minor issues:

Please review the references throughout the text. For instance, missing reference in line 1022.

We have revised all the references and added the missing citation in the legend of Fig ED12 (which was line 1022 before the revisions now line 1151).

Reviewer #4 (Remarks to the Author):

In the manuscript "A mitoribosome with 53 rRNA molecules requires extensive protein and RNA repurposing to function" by Shikha et al., the authors present a high-resolution structure of the *Toxoplasma gondii* mitoribosome using cryoEM and RNA sequencing technologies. They determined the 3D structures of all 53 rRNAs, revealing that the rRNAs in the mitoribosome core possess poly-A tails. The enzyme responsible for this polyadenylation was identified as TgmtPAP1. The study highlights the significance of rRNA polyadenylation, protein extensions, and proteins with transcription factor-like domains in the formation and stability of the mitoribosome. This work represents the first successful resolution of the *T. gondii* mitoribosome structure, marking a significant advancement in the field of *T. gondii* research.

Major Points:

1. Please include quantification for both plaque size and number in all plaque assays.

Since we get no plaques or extremely small plaques upon depletion of these proteins, quantification is technically challenging. However, in the previous round of revisions reviewer 1 (who was replaced for this round with the new reviewer 1) have already asked for a quantitative way to measure the growth defect. In response to that comment, we have included the replication assay in Fig 3B.

2. Please add quantification of all western blots. It seems that the bands for Complex V in Figure 1E appear overexposed. Please verify that no overexposed blots are presented, as this can lead to inaccurate quantifications.

For the TgmtPAP1 KD in Figure 1E and mL161 KD in Figure ED11C, it appears that normalizing the bands for Complex V to the bands for TgTom40 may show a decreased level of Complex V after the knockdowns. Please include quantification to verify this observation. If there is a decrease, please discuss how this might affect Complex V, given that the *T. gondii* mitoribosome is not reported to translate proteins for Complex V.

To address all these concerns regarding over-expression and method of quantification and calculation of the presented Complex V bands, we now provide quantification of three independent experiments performed for all four cell-lines, whereby we normalised the signal from complex V to the signal from TgTom40 as the reviewer suggested and showed that there is no significant change in the CV signal. This is provided in the revised Fig ED11.

3. Can the author specifically state in the text which three proteins involved in the electron transport chain are translated by that *T. gondii* mitoribosome? Are they COX1, COX3 and COB? COX1 and COX3 are components of Complex IV, and COB is part of Complex III. When TgmtPAP1 or other mitoribosome proteins are knocked down, the authors demonstrated that the assembly and activity of complex IV are disrupted using functional assays, while Complex II activity, indicated by SDH activity assay, remains unaffected. Could the authors include an assay to evaluate the assembly and activity of complex III?

As the reviewer suggested, we have mentioned the three proteins that are encoded by mitochondrial genome in the revised text (lines 190) and we explained that our assay used complex IV as an example of one of the two complexes that depend on the mitoribosome for assembly (which accounts for two of the three proteins encoded in this genome, 66%). These data provide support for a translation defect, which is presented in addition to a defect in mitoribosome assembly. While we can see that it would be more thorough to include also complex III analysis, it is not necessary to make the point about mitochondrial translation defect, as we published in the past (PMID: 35630308; PMID: 31339607). We should explain that we did try (very hard) to add the requested experiment in two ways: (1) Complex III activity assay. This assay was not performed in our lab before (and only published for *Toxoplasma* in one publication, thus not a routine in the field). 10 attempts technically failed in our hands (the atovaquone control doesn't differ from the untreated line, meaning there is background signal that we are not able to clear. Below 4 examples). We further tried to create a line where a complex III component is tagged in the knock-down background so that we might be able to follow the complex assembly upon down-regulation and this failed too (we made 6 x independent transfections aiming to recreated the TgmtPAP1 knock-down in a complex III tag background, and from 4 x positive polyclonal pools and an average of 40 clones tested per pool, all clones were negative. We tried a further 3 x independent transfections for tagging a complex III subunit in the background line of our TgmtPAP1 knock-down – only 1 x positive polyclonal pool was obtained, and 110 clones tested – all negative). As stated, we still believe that the presented data supports our conclusions, however we now clarify the reviewers' point in the revised text (line 196-201).

4. How can you ensure that the rRNA fragmentation observed in the cryoEM structure and RNA sequencing data is not an artifact of the isolation process? Did you use any controls to rule this out?

As a point of clarification, we wish to highlight that the rRNA is already encoded in fragments in the mitochondrial genome. Thus, the fragmentation is supported to be a genuine biological feature by three separate and independent experiments: (1) the mitochondrial genome sequence (performed by others: PMID: 33906963 PMID: 38363119); (2) the sequences that were identified here directly from the density (and which therefore occurred in >250,000 particles); and (3) the RNA sequencing we reporter here. Specifically, our ability to map most of the fragments that we found directly from the structure (prior to performing the RNA sequencing as a validation) against the available genome sequences (Table S4 and Fig ED5) provide support to their authenticity.

We revised this in the text (line 110) to clarify that the experiments are independent.

We cannot fully exclude that the 9 fragments (out of 53) not found in the genome did not result from fragmentation during the preparation, however the presence of novel proteins that evolved in these parasites to bind and stabilise these small fragments is in support of their authenticity too. Still, we discuss this caveat in the revised discussion to be fully unbiased (lines 411).

5. For the mass spectrometry analysis, the data was searched only against the ToxoDB database. Since the parasites were grown in Vero cells (host cells), there could be contamination from host proteins. If the mitoribosome proteins from Vero cells are not drastically different from those of the parasites, their electron density in the cryoEM analysis could be indistinguishable. Please analyze the mass spectrometry data against a protein database for Vero cells to determine if there is any contamination from the host cells.

A scenario whereby host ribosomal proteins make their way through the parasitophorous vacuole, the parasite plasma membrane, and the parasite mitochondrial membrane to integrate into the parasite mitogenome is extremely unlikely. Moreover, we would like to reiterate that the identity of the proteins found in the parasite mitoribosome structures was directly deduced from the density (which is based on >250,000 homogenous particles), so we have full certainty that these are all *Toxoplasma* proteins.

To address the reviewer comment and for full transparency, we now provide the analysis of the mass spectrometry data against Vero genome, this new analysis is added as a new tub within the revised Table S1. As expected, many of the peptides found do “hit” Vero mitochondrial proteins. This is expected for two reasons: (1) in the many cases of proteins that share high similarity between parasite and host, the peptides from the *Toxoplasma* proteins also match the Vero homolog; (2) we may well have some host mitochondria contaminants in the preparation that are not part of the mitoribosome but that are still picked-up in the mass spectrometry. But, the mass spectrometry data was merely used to confirm the proteins already directly identified from the density unambiguously recognising the *Toxoplasma* proteins.

Minor points:

1. Figure 1: when labeling mRNA entrance and tunnel exit, please use a line or arrow to indicate their locations, so that the labels and their backgrounds do not obscure the structure.

We have revised Fig 1 to include the lines as suggested by the reviewer.

2. Line 255: It is not accurate to state that “Several transcription factors are....” As the author stated in the discussion, “it is unlikely that these proteins simultaneously serve as transcription factors, as they lack other functional domains such as the AT hook, zinc-finger, ACDC often found in AP2 transcription factors.” It would be more accurate to say, “Proteins contain transcription factor-like domains are....”

We have changed this sub-title to “Several proteins with transcription factor domains are repurposed to compensate for a reduced subunit interface.”

3. Lines 264 and 265: Please add an explanation why mL151 has a BoIA-like domain, and mL161 has an ApiAP2-like domain. Do mL151 and mL161 have DNA-binding-like domains or domains that mediate transcription?

The ApiAP2 of mL161 was found via BLAST that identified its' *Plasmodium* orthologue which is published as an ApiAP2 member (PF2D7_0934400, PMID: 30959972). We have added this paper reference in the revised text. Likewise, mL151 BoIA-like domain was identified via foldseq and DALI, based on structure-structure comparison, we have explained this in the revised text.

4. Line 410: In this paragraph, the authors discuss targeting the *T. gondii* mitoribosome for drug development. The paper compares the differences between *T. gondii* mitoribosome and bacterial mitoribosome but does not mention differences between *T. gondii* and human mitoribosomes. Please consider adding a description of why the *T. gondii* mitoribosome might be different enough from the human mitoribosome to serve as drug targets.

The differences between the bacteria ribosome and *Toxoplasma* mitoribosomes discussed in the original submission provides an example of how all mitoribosome have some differences from one another which could potentially underpin differential sensitivity to inhibitors. As the reviewer suggested we added to the discussion about the potential of apicomplexan mitoribosomes as a drug target.

Reviewer #5 (Remarks to the Author):
